# Micro-costing from healthcare professional's perspective and acceptability of cutaneous leishmaniasis diagnostic tools in Morocco: A mixed-methods study

Issam Bennis[1]*, Mohamed Sadiki[2], Abdelkacem Ezzahidi[3], Naoual Laaroussi[4], Souad Bouhout[4]

1 International School of Public Health, Mohammed VI University of Health and Sciences (UM6SS), Casablanca, Morocco, 2 Delegation of Ministry of Health, Tinghir Province, Morocco, 3 Delegation of Ministry of Health, Ouarzazates Province, Morocco, 4 Directorate of Epidemiology and Diseases Control, Ministry of Health, Rabat, Morocco

* issambennis@gmail.com

## Abstract

In Morocco, cutaneous leishmaniasis (CL) represents a concern with three causative parasite species. Despite this, rapid diagnostic test (RDT) for CL is not part of the national control strategy. This study evaluates the acceptability and micro-costing of the CL Detect Rapid Test by Inbios International. The study was conducted from June 2019 to January 2020 and included 46 healthcare professionals from 40 primary healthcare centers and district labs. Data was collected by self-administered questionnaires and interviews and analysed using NVivo, Jamovi, and Python to generate a predictive model and sensitivity analysis by calculating the average Cost-Benefit Ratio for compared CL diagnostic scenarios. The exchange rate is 1 USD = 9.6 MAD (Moroccan Dirham) in November 2019. The CL-RDT received notable acceptance for its user-friendliness and time efficiency compared to traditional microscopy. Micro-costing data revealed that the average unit cost for microscopy is 15 MAD [7–31], whereas 75 MAD [52–131] for CL-RDT. Altogether, the diagnostic cost for microscopy is 115 MAD±4, marginally higher than the 102 MAD±2 for CL-RDT (p = 0,05). Sensitivity analysis identified the most cost-benefit scenarios based on a Cost-Benefit Ratio (CBR). The optimal approach involves using CL-RDT once at a primary healthcare centre (PHC) (CBR = 1.4), especially if the unitary cost is below 79 MAD. The second-best option is using CL-RDT once at a laboratory (CBR = 1.0), which is advantageous if priced under 54 MAD. However, using CL-RDT twice for the same lesion had a less favourable CBR of 0.6 and was only beneficial if priced below 09 MAD. The reference scenario was a single CL-RDT at the PHC followed by microscopy at a laboratory. In conclusion, the forthcoming CL-RDT, expected to feature enhanced sensitivity, is advocated for deployment in resource-limited endemic areas.

**Data Availability Statement:** In accordance with the PLOS Data Policy, we confirm that all anonymized data underlying the findings are available within the manuscript and its supplementary information files. Due to ethical considerations, the complete raw data set cannot be made publicly accessible, but anonymized data are provided to support the study's conclusions.

**Funding:** This work was supported by the TDR-SGS for Implementation Research in Infectious Diseases of Poverty coordinated by WHO/EMRO/TDR Grant number (SGS 18-63) Reference 2019/893794-1. The funder had no role in study design, data collection and analysis, decision to publish, or preparation of the manuscript.

**Competing interests:** The authors have declared that no competing interests exist.

# Background

In north Africa, cutaneous leishmaniasis is a skin disease caused by the Leishmania parasite, which results in permanent scars on visible body parts that are more damaging psychologically for women [1–3]. Factors such as the size and location of lesions, treatment decisions, and treatment response influence the epithelialisation process, ultimately determining whether or not lasting visible scars form [1,4]. Prompt detection and diagnosis of Cutaneous Leishmaniasis during the initial weeks could facilitate timely treatment decisions and improve aesthetic skin outcomes [5].

A recent rapid diagnostic test for cutaneous leishmaniasis, CL Detect Rapid Test, has been developed by Inbios International Seattle, USA [6]. This test identifies the peroxidoxin antigen on the amastigote membrane of the parasite. Healthcare professionals (HP) can quickly obtain a sample using a small dental broach on a clinically suggestive ulcerative lesion, and the test can be performed during the first patient visit in under 40 minutes without additional lab tests (Witness Video demonstration in **S1 Video**).

In Eastern Mediterranean countries where Cutaneous Leishmaniasis is endemic, the standard laboratory test involves light microscopy examination of a smear prepared by scraping a suspected lesion. This method enables the visual identification of parasites following Giemsa stain colouration [7]. Ideally, results are obtained in approximately one and a half hours when colouration and microscopy readings are performed directly [8]. However, in many remote areas, in Morocco, where CL are endemic with each five years cyclicity for ZCL and almost a sustainable yearly trend of ACL cases (**S1 Table** & **S1 Fig**), people rely for CL diagnostics on primary health centres (PHCs). Then, patients often require two to three PHCs and lab visits over several days to receive microscopy results.

The Moroccan healthcare system for CL management employs a pyramidal network of primary healthcare facilities, ensuring equal access in urban and rural areas. Initial examinations of suspected CL cases predominantly use microscopic testing for diagnosis, a widely implemented and fundamental method. Although the potential of CL rapid diagnostic tests (RDTs) could be acknowledged, their integration into the system is still pending, making microscopy the primary diagnostic tool in use [9].

The CL-RDT was evaluated in a phase III diagnosis trial between 2016 and 2017 and 2022 by three separate research teams in three countries, targeting *Leishmania tropica* species in Afghanistan, *L. tropica*, *L. major, and L. infantum* species in Morocco, and *L. tropica*, *L. major* in Iran [9–11]. The results exhibited a sensitivity (Se) of 65 to 79% and a specificity (Sp) of 90 to 100% compared to microscopy (Se 65%—Sp 100%) [9]. Its performance compared to microscopy was slightly less sensitive than PCR tools as the gold standard [9–11]. A recent study from Peru using modified sampling scraping techniques found CL Detect Rapid Test sensitivity compared to microscopy with (64%) scenarios where the sample was collected using a dental broach and became (83%) when the sample was collected by Lancet [12]. However, in positive subjects confirmed by PCR for CL Detect Rapid Test and collected by scraping, microscopy had 08 false negative cases 108 [12]. According to other information from Ethiopia, CL-RDT was almost 10% more sensitive when using lancet scarping than dental broach sampling.

The same previous study in Morocco has shown that microscopy requires well-equipped laboratories with trained technicians and is a time-consuming process compared to the Rapid Diagnostic Test (RDT) with a recommendation to do microscopy in case of negative RDT tests conforming to the exigences of Phase III diagnostic trials studies [9]. Consequently, a quick diagnostic method for CL, such as the RDT, would considerably enhance patient comfort. However, experience with malaria RDTs indicated that limited health system capacity

and socio-economic, political, and historical factors hindered their implementation at primary healthcare facilities, with healthcare professionals (HPs) reluctant to add more tasks to their workload [13]. Concerns also arose about the test's accuracy in low-endemic areas, notably when it was the sole basis for therapeutic decisions. Additionally, community health workers expressed the need for training and regular supervision, while challenges were encountered with patients' transportation and diagnostic preferences [14–16].

As this CL-RDT is statistically more accurate than microscopy in the context of *L.tropica* and *L.major* the two main species present in CL endemic areas in Morocco, integrating it into health strategies requires a thorough understanding of the acceptability and cost implications of this new tool from healthcare professionals' perspective by evaluating their experiences with the Cutaneous Leishmaniasis diagnostic process and interactions with patients and their families during daily practice. In addition, defining the Cost-Benefit Ratio is a crucial metric that helps policymakers assess the economic viability of different intervention scenarios.

Therefore, this study aims to fulfil two objectives: Firstly, to quantitatively assess the micro-costing of CL diagnostic tools among healthcare professionals in endemic regions of Morocco. Secondly, to document the acceptability and advantages of using either CL Rapid or microscopy for diagnosis.

## Material and methods

This explanatory mixed methods study uses a comparative analysis approach to investigate the micro-costing and acceptability of Cutaneous Leishmaniasis diagnostic tools in Morocco from the perspective of healthcare professionals.

### Population and study settings

A stratified sampling approach was used to select a representative number of health facilities where HPs manage CL cases in four CL-endemic Moroccan provinces (Errachidia, Ouarzazate, Tinghir, and Sefrou). The sampling included all high-endemic PHCs within the previous three years in the four Moroccan provinces, covering areas predominantly affected by *L. major or L. tropica*. The study involved 35 primary health centres (PHC) and five district laboratories, with all patient information obtained from medical records being anonymised. The analysis unit in this study is the Healthcare professionals.

### Data collection and sampling strategy

Between 26 August 2019 and 28 January 2020, healthcare professionals (HPs) who frequently managed CL patients at each visited primary health centre and district laboratory were invited for face-to-face audio-recorded individual in-depth interviews (46 HPs in total) (**S2 Table**).

The principal investigators' team included two male doctors and two male administrative nurses experienced in participating in previous scientific studies. Each investigator worked in one of the four selected provinces. They received a full-time, one-week training workshop on qualitative interviews, cost-effectiveness, and an overview of the project objectives, rationale, research questions, and data collection tools. Each investigator respected the confidentiality criteria, and ethical consent was obtained before starting the video demonstration and the interview record. Structured face-to-face interviews were conducted using a topic guide organised by thematic questions (**S1 Text**). Then, initial interview discussions involved 40 HPs, some with prior experience using the CL-RDT, some knowledgeable about it, and others unaware of its existence. All participants watched a video-recorded demonstration explaining the use of the CL-RDT (as shown in **S1 Video**) before answering the main interview questions.

Six additional interviews were conducted by the principal investigator (PI) using another topic guide (**S2 Text**) to explore emerging ideas further, support quantitative responses, and achieve saturation. These interviews targeted six new participants with years of experience in CL-endemic areas.

The PI recorded information from the qualitative questions in a notebook linked to each participant's thoughts and responses.

The principal investigators' team revisited the 40 initial HPs to check the self-administered quantitative questionnaire. This questionnaire collected clinical information on past CL patients and data on costs associated with CL diagnostic confirmation concerning overall time and CL diagnostic pathways from the HP perspective (**S3 Text**).

Diagnostic tool costs were estimated using a micro-costing strategy, which involves listing the overall costs of each input used for diagnosing a specific disease (including staff time, supplies, and out-of-pocket expenses) [17]. This strategy is suitable for new intervention costing and capturing specificities and complexities in activities that might be present in different healthcare settings based on the HPs' perspective [18]. In our study, the micro-costing included the direct enumeration and costing of all inputs consumed for patient diagnostic and program implementation. These costs were then correlated with unit cost data from the 40 primary healthcare facilities, including centres and district laboratories, where the study's health professionals worked [19]. The exchange rate used was 1 USD = 9.6 MAD (Moroccan Dirham) as of November 2019.

In-depth information was gathered on every resource used in the CL diagnostic process. The unit cost of each resource included direct costs such as the cost of the CL-RDT or microscopy tests, the time spent by healthcare professionals on each test, and the cost of any consumables used. It also covered indirect costs like patient travel and the overhead costs associated with using the facilities.

In the pursuit of standardisation and transparency in micro-costing for cutaneous leishmaniasis (CL) diagnosis from a healthcare professional's perspective, our study delineates costs into fixed, variable, and combined categories [17].

<u>Fixed costs</u> refer to expenses that do not change with the activity level within a specific range. In this study, fixed costs include:

- Equipment and laboratory material costs: These are the expenditures for acquiring, maintaining, and using equipment and materials necessary for CL diagnosis in laboratories.

- Salaries of healthcare professionals: Regular wages for medical staff, including laboratory technicians, physicians, and nurses, independent of the volume of CL tests conducted.

- HP working time: The financial equivalent of healthcare professionals' time devoted to CL diagnosis, which includes their fixed salaries.

- Distance costs to reach facilities: These cover the travel-related expenses incurred when patients or samples must be transported to diagnostic centres.

- Driver transfer: The cost of transport services to move patients or samples to and from the healthcare facilities.

- Patient self-transfer: Out-of-pocket expenses for patients travelling to healthcare facilities for diagnosis.

- Administrative MoH costs: Operational costs handled by the Ministry of Health for the administrative side of CL diagnostics, including data handling and record-keeping. In our framework, these administrative costs are not charged to the patient.

 

Variable costs change with the level of activity. Key variable costs in this study are:

- Time testing: The duration-dependent cost associated with performing the CL diagnostic test.

- Diagnostic strip or slide unit purchased: The expense of individual diagnostic strips for RDT or microscopy slides is essential for the CL testing process.

- Additional transport costs: Extra travel costs related to the diagnostic procedure, potentially necessitated by additional testing or the inadequacy of initial diagnostic tools.

Combined costs: Integrating fixed and variable costs is pivotal in comprehensive micro-costing analysis [18].

- Transportation costs: An aggregation of all travel expenses, encompassing distances to facilities and additional costs, which fluctuate based on the diagnostic tool used.

- Workload diagnostic costs: Costs impacted by the complexity of tool manipulation and the time investment by healthcare staff in the diagnostic process.

- Diagnostic confirmation scenarios represent the potential additional costs incurred when further testing is needed to confirm CL diagnoses, influencing the total costs.

- Overall CL diagnostic costs: This is the cumulative impact of the cost categories from the initial patient interaction to the final diagnosis confirmation.

The cost categorisation flowchart for CL diagnostic tools from a Healthcare professional's perspective is presented in **Fig 1**.

Particular emphasis was placed on gathering data until a point of saturation was reached, i.e., until additional data did not significantly change the overall cost estimates. Pilot data was initially collected from a small sample to estimate the range and standard deviation of costs.

Detailed records were maintained for every diagnostic test, whether CL-RDT or microscopy, to capture the entire cost structure accurately.

## Data analysis

For the quantitative section, two people recorded participant characteristics twice in a Microsoft Excel sheet (a Biology master's student freelancer not involved in the study and the PI). After comparing and cleaning the raw data, the answers were categorised as binary numeric or dichotomous variables.

The statistical analysis included micro-costing using Excel (Microsoft Corporation) to record structured needed data. The corresponding descriptive statistics for tables and the linear regression predictive model use Jamovi software (version 2.3.24), an open-source platform that offers a comprehensive suite of statistical tools, ensuring rigour and reproducibility in the analysis.

A sensitivity analysis was performed using a powerful programming language, Python (version 3.11.1) [Python Software Foundation. Available at http://www.python.org], targeting the Sobol method facilitated by SALib library, a free, open-source Python library designed to conduct sensitivity analyses, providing a range of algorithms. The Sobol method is a global sensitivity analysis technique used to investigate the influence and importance of input variables on output (or set of outputs) to understand how uncertainties in the inputs affect the outputs of a model. The Sobol method can assist researchers and policymakers in focusing their attention on the most critical variables, facilitating more effective decision-making. The data were processed and visualised in Jupyter Notebooks [20].

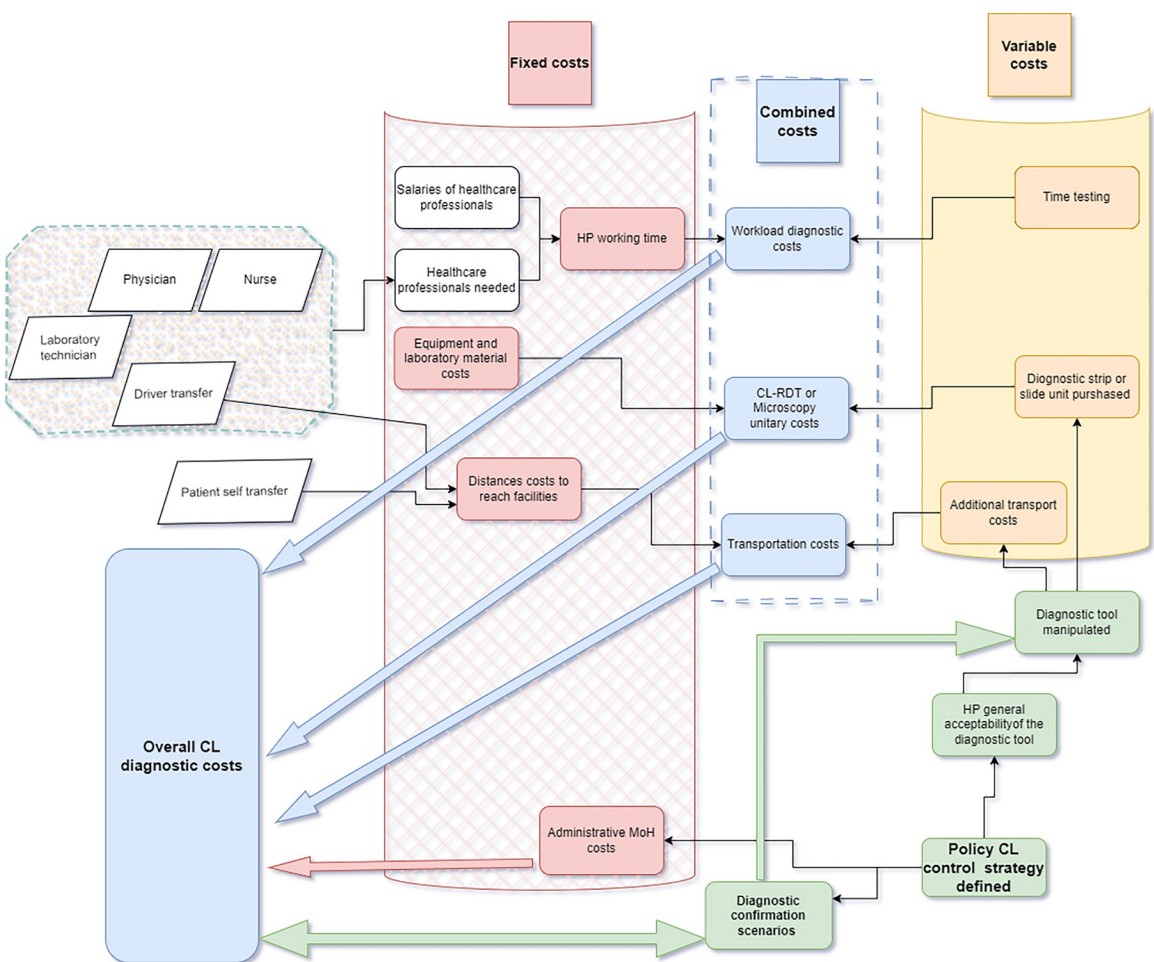

**Fig 1. Cost categorisation flowchart for CL diagnostic tools from a healthcare professional's perspective.** The green part of the figure includes: Healthcare Professional (HP) General Acceptability of the Diagnostic Tool: This element suggests that the overall usage and cost-effectiveness of a diagnostic tool for CL are heavily influenced by how well healthcare professionals accept it. If the tool is widely accepted, it will likely be used more frequently, thus affecting the variable costs associated with diagnostics. Policy CL Control Strategy Defined: This signifies the strategic decisions made at a policy level which affect the overall costs of CL diagnostics. A well-defined control strategy can streamline the diagnostic process, potentially leading to more efficient resource allocation and reduced overall costs. Diagnostic Confirmation Scenarios: These are specific cases or circumstances under which the diagnosis of CL is confirmed. The scenarios chosen for confirmation can impact the fixed and variable costs due to the different resources and time required for each scenario.

This supplementary sensitivity analysis was conducted to investigate how changes in costs of the unitary cost of the RDT and other independent variables, such as the diagnostic workload cost, could influence the threshold of the overall cost of CL Rapid Diagnostic Test (RDT) compared to the the the microscopy's overall cost. Indeed, those independent variables were systematically varied within a reasonable discount rate (lower to upper limits). This range was chosen to account for potential uncertainties and to provide a comprehensive understanding of how the choice of discount rate impacts the study's conclusions. Possible real-world scenarios were defined to determine how parameter changes affected the cost of diagnosing cutaneous leishmaniasis by calculating the cost-benefit rates.

The main assumptions in this analysis centred first on the fixed expenses for the supplies needed in the diagnostic process. Second, the HPs' salaries were considered based on their responses. The cost of workload time spent on CL diagnostics was calculated based on 8880

minutes of regular duty work per month in the public sector and the time required for their actual contribution to the diagnostic process, either with or without supplementary additional delay times in obtaining the results. Consequently, we could understand the variables with the most significant influence on the cost.

For the qualitative section, a thematic analysis was conducted on the qualitative data to primarily examine the acceptability of introducing the CL-RDT at primary health facilities and the motivating factors for HPs to use the CL-RDT based on their previous experience with CL.

The analysis focused on the observed and expected patient pathways when using the CL-RDT as a diagnostic tool, as reported by the HPs. After transcribing and translating the audio-recorded interviews, the principal investigator employed two methods to improve the reliability of the research analysis. In the first phase, the one sheet of paper (OSOP) qualitative technique was used to categorise themes emerging from the quotes by placing similar ideas in the exact location on a large sheet of paper to help explain the data [21]. In the second phase, NVivo 2020 software was used to analyse interactions and relationships qualitatively, allowing for easy triangulation of the main themes with the quantitative data by matching the same interviewee with the same questionnaire responses. The findings from both phases highlighted differences in overall acceptability levels among HPs based on their previous or new knowledge of the CL-RDT.

The qualitative coding tree developed from the text data was triangulated with the average information collected from the quantitative section and the micro-costing analysis. The triangulation enhanced the information on the final assumptions about the time required to perform the diagnostics (by calculating the overall workload for each diagnostic tool, the accuracy of the test and the acceptability of six different scenarios that could be adopted as CL diagnostic strategy in Morocco. Then, the average Cost-Benefit Ratio (CRB) could calculate the scenario with the highest accuracy level (Scenario 3: RDT at the PHC followed by microscopy at the laboratory for confirmation).

This CBR had the following assumptions of composite variables of test accuracy calculation, test acceptability perception, saving patient time perception and laboratory reliability perception, all possible to define based on the previous study and the triangulation of the qualitative and quantitative data of the current study. The normalised Average CBR for each scenario is a quotient of its benefits and the benefits of the reference scenario.

$$Normlaized\ CBR = \frac{Benefit\ of\ the\ scenario}{Cost\ of\ the\ scenario} \div \frac{Benefit\ of\ the\ reference\ scenario}{Cost\ of\ the\ reference\ scenario}$$

The reference scenario has CRB = 1. A higher CBR indicates a more favourable cost-benefit profile, suggesting that the diagnostic strategy provides better value for the money spent.

The essential Python script used in the analysis is available in (**S1 Codes**).

## Ethical considerations and good publication practices

All HPs provided formal written consent to participate in the study, including informal discussions outside the audio-recorded interview process. The Ethical Committee of Biomedical Research of the Faculty of Medicine in Rabat approved the study. Authorisation number 36/2019 was granted in March 2019 and carried out with an official Ministry of Health (MoH) National authorisation. The study protocol included the economic evaluation plan and the modified questionnaire based on the previously published cost-effectiveness study [22].

The presentation of the qualitative section adheres to the Standards for Reporting Qualitative Research (SRQR) checklist (**S4 Table**) [23]. The production of the quantitative section

**Table 1. General characteristics of the participating facilities based on provinces, types, data collection, and healthcare professionals' occupations and gender.**

| Province localisation | CL species area dominance | Questionnaire Participating facilities | | Interview types and gender of participants (Man vs. Woman) | | | | Healthcare professional occupation | | |
|---|---|---|---|---|---|---|---|---|---|---|
| | | Primary Health Centre | District Laboratory | Man, initial Interviews | Woman initial interviews | Additional In-depth Interviews | All Interviews | Nurse | Tech Laboratory | Doctor |
| Errachidia | *Leishmania major* (21) | 12 | 01 | 7M | 6W | 1M + 3W | 17 | 14* | 1 | 2 |
| Ouarzazate | | 07 | 01 | 6M | 2W | 1M + 1W | 10 | 7* | 0 | 3* |
| Tinghir | *Leishmania tropica* (19) | 12 | 02 | 7M | 7W | 0 | 14 | 12 | 2 | 0 |
| Sefrou | | 04 | 01 | 4M | 1W | 0 | 5 | 4 | 0 | 1 |
| Total | | 40 | 35 | 05 | 24M | 16W | 2M + 4W | 46 | 37 | 3 | 6 |

(*) Focused interview (FI) 42, 43, 44, 45 from Errachidia (E) and FI 14, 46 from Ouarzazate (Z) participated as the final six focused In-depth interviewees to reach saturation.

The principal investigator did those interviews.

attaches to the Consolidated Health Economic Evaluation Reporting Standards (CHEERS 2022) checklist (**S5 Table**) [24].

## Results

In the study, six doctors (average age 33), three laboratory technicians (average age 48), and 37 nurses (average age 43) participated in the quantitative data collection and either initial face-to-face interviews or additional final qualitative interviews. The Ministry of Health employed all participants. The gender distribution included 26 men and 20 women (**Table 1**).

### The quantitative results linked to CL diagnostic

**Costs categorization.** As shown in **Table 2**, which includes the main results of the cost catégorisation values comparing the microscopy to the CL-RDT, the cost analysis for diagnosing cutaneous leishmaniasis in Morocco using microscopy and CL-RDT, while initially calculated under the assumption of one lesion per patient, needs adjustment considering the actual lesion count. Both methods start with identical equipment, and asepsis costs 10 MAD for a single lesion. Additional consumables for CL-RDT bring its specific costs to about 3 MAD more. Microscopy slides are cheaper at 5 MAD compared to 62 MAD for a CL-RDT strip. The overall unitary cost is lower for microscopy at 15 MAD than 75 MAD for CL-RDT. However, CL-RDT shows a significant cost advantage in workload, averaging 27 MAD, much lower than microscopy's 99 MAD. This extends to overall diagnostic costs (100 MAD for CL-RDT vs. 115 MAD for Microscopy) and transport costs, which are lower for CL-RDT. Administrative costs from the Ministry of Health are assumed to be zero for both methods.

**Average of CL lesion number on the cost analysis of CL-RDT and microscopy.** The cost analysis of CL-RDT and microscopy is appreciably affected by the mean number of CL lesions per patient, with a noted range from 1.7 in Tinghir to 2.2 in Errachidia, Ouarzazate, and Sefrou, as indicated in **S1 Table**. Such variability implies a potential escalation in actual costs from the base figures, predominantly for CL-RDT. Theoretically, considering the average lesion count, the per-patient cost for CL-RDT might increase commensurately, influencing its cost-effectiveness relative to microscopy. For example, in localities with a 2.2 lesion average, the diagnostic expenses for CL-RDT could surpass double the projected cost for a singular lesion. Nonetheless, practical clinical approaches, as qualitatively described, involve selecting

**Table 2. Detailed micro-costing of microscopy and rapid diagnostic test for cutaneous leishmaniasis from a healthcare professional's perspective.**

| Costs Categories | Cost value/Range for Microscopy | Cost value/Range for CL-RDT | Notes |
|---|---|---|---|
| One lesion sample equipment protection and asepsis cost | 10 MAD [4,6–22] | 10 MAD [4,6–22] | Necessary consumables for diagnosing one lesion are similar to CL microscopy and CL-RDT. Detailed descriptions are available in S3 Table* |
| Additional dental broach, reaction cup and pipette tip and tube for CL-RDT cost | | 3 MAD [1,6–7] | The study defines consumables within the year of study; for specifics, refer to S3 Table * |
| Microscopy unit slide cost | 5 MAD [2–9] | | The study defines consumables within the year of study; for specifics, refer to S3 Table * |
| CL-RDT unit strip cost | | 62 MAD [46–102] | The study defines consumables within the year of study; for specifics, refer to S3 Table * |
| Microscopy unitary cost | 15 MAD [7–31] | | Cost calculations include the addition of the necessary consumables as outlined, with lines 1 and 3 being pertinent to different diagnostic tools. |
| CL-RDT unitary cost | | 75 MAD [52–131] | Cost calculations include the addition of the necessary consumables as outlined, with lines 1, 2, and 4 pertinent to different diagnostic tools. |
| Workload costs | 99 MAD [42–276] | 27 MAD [7–67] | Costs are partly based on the healthcare professionals' time required to perform the diagnostic process without accounting for transfer or transport delays, noting that microscopy is typically more expensive and time-consuming than CL-RDT (as detailed in Table 4). |
| Overall Diagnostic Cost | 115 MAD [58–292] | 100 MAD [79–131]** | The aggregated costs combine fixed and variable elements, including the workload and unitary equipment costs, and vary according to the unitary cost of the RDT strip (refer to Table 4 for scenario-specific costs). |
| Average transport cost | 122 MAD [8–360] | 67 MAD [6–190] | Geographical distance-related expenses and additional transport costs depend on various assumptions and scenarios, fully described in Table 5. |
| Maximal transport cost | 224 MAD [8–720] | 124 MAD [8–380] | Geographical distance-related expenses and additional transport costs depend on various assumptions and scenarios, fully described in Table 5. |
| Administrative MoH cost | 0 | 0 | Administrative costs, as determined by the Ministry of Health regulations, are assumed to be free of charge in this context. |

(*) S3 Table lists the additional equipment and laboratory materials needed for microscopy and RDT CL diagnostics.

(**) if the unitary strip cost is 62 MAD (1 USD = 9,6 MAD).

the most representative lesion for sampling and reserving additional lesion sampling for cases with initial negative results.

**Comparative analysis of time, workload, and transport costs in CL diagnostics: CL-RDT vs. microscopy.** Table 3 emphasises the differences in time needed to manipulate CL diagnostic tools, with the average time for immediate initial manipulation being 28 minutes (95% CI: 26 to 31) for CL Detect Rapid Test and 49 minutes for microscopy. The average information delay experienced by the initial requester of the CL diagnostic exam to get the result, based on various patient transfer or sample transport scenarios, varied significantly, with 40 minutes (95% CI: 18 to 62) for CL Detect Rapid Test compared to 2486 minutes (≈41 hours) for microscopy. Other metrics also showed significant differences, consistently indicating that the CL Detect Rapid Test required less time across various scenarios, especially without considering the transfer or the transport time CL-RDT remained faster with 41 minutes compared to 155 minutes for microscopy (p<0,001).

Table 4 details the costs associated with CL diagnostic tools, revealing that the microscopy CL workload diagnostic cost averaged 99±43, while the CL-RDT workload diagnostic cost averaged 27±21, with a significant p-value of 0.009. Additionally, even if the microscopy unitary cost was 15 MAD, the overall microscopy CL diagnostic cost becomes 115 MAD±43,

**Table 3. Time needed to manipulate CL diagnostic tools based on the different transfer or transport delay scenarios and HP availabilities.**

| N = 40 (RDT = 6) | Diagnostic tool manipulated | Mean (in minutes) | 95% confidence interval | | | *p-value |
|---|---|---|---|---|---|---|
| | | | Min | Max | Median | |
| Time for immediate initial manipulation of the RDT or microscopy | CL Detect Rapid Test | 28 | 25 | 31 | 30 | **0.016** |
| | Microscopy | 49 | 32 | 65 | 48 | |
| Average time with information delay of the diagnostic result | CL Detect Rapid Test | 40 | 18 | 62 | 33 | < .001 |
| | Microscopy | 2486 | 1431 | 3540 | 1440 | |
| Minimal delay time for information on the diagnostic result | CL Detect Rapid Test | 40 | 19 | 62 | 33 | < .001 |
| | Microscopy | 1607 | 789 | 2424 | 1440 | |
| Maximal delay time for information on the diagnostic result | CL Detect Rapid Test | 45 | 25 | 65 | 40 | < .001 |
| | Microscopy | 4394 | 2697 | 6091 | 4320 | |
| HP working time to perform the diagnostic process without transfer or transport delay | CL Detect Rapid Test | 40 | 19 | 62 | 33 | < .001 |
| | Microscopy | 155 | 120 | 189 | 130 | |

(*) *p-value* calculated by ANAOVA unidirectional (Welch test) as Normality and homoscedasticity conditions were not found.

instead of CL-RDT overall cost linked to the percentage of the unitary strip cost. For example, it ranges from 83±21 to 100±21 if the unitary strip is 45 or 62 MAD, respectively.

The p-values obtained from a unidirectional ANOVA with Fisher's classic test indicate the statistical significance of the differences in costs associated with the diagnostic tools, which can influence healthcare professionals' (HP) preferences or acceptability of these tools. The results

**Table 4. Descriptive analysis of workload and diagnostic tools costs for CL-RDT compared to microscopy.**

| | N* | Mean | SD | Min | Max | p-value ** |
|---|---|---|---|---|---|---|
| CL microscopy workload diagnostic cost (based on HP working time to perform the diagnostic process without transfer or transport delay) | 34 | **99** | 43 | 42 | 276 | |
| CL-RDT workload diagnostic cost (based on HP working time to perform the diagnostic process without transfer or transport delay) | 6 | **27** | 21 | 7 | 67 | **0.009** |
| Microscopy overall CL cost (workload + the unitary equipment and laboratory material costs) | 34 | **115** | 43 | 58 | 292 | |
| CL-RDT overall cost (workload + the unitary equipment and laboratory material costs) if the unitary strip cost is equal to 45 MAD | 6 | 83 | 21 | 62 | 122 | **0.02** |
| CL-RDT overall cost (workload + the unitary equipment and laboratory material costs) if the unitary strip cost is equal to 54 MAD | 6 | 92 | 21 | 71 | 131 | **0.04** |
| CL-RDT overall cost (workload + the unitary equipment and laboratory material costs) if the unitary strip cost is equal to 62 MAD | 6 | 100 | 21 | 79 | 139 | 0.05 |
| CL-RDT overall cost (workload + the unitary equipment and laboratory material costs) if the unitary strip cost is equal to 74 MAD | 6 | 112 | 21 | 91 | 151 | 0.09 |

(*) It indicates that among the 40 healthcare professionals participating in the study, six had prior hands-on experience with the CL-RDT. These experienced individuals were from six distinct healthcare facilities, as elaborated in S2 Table. Conversely, the remaining 34 participants were exclusively acquainted with the CL-RDT procedure through a video demonstration.

(**) The p-value was calculated by ANOVA unidirectional with the Fisher classic test due to respect for normality and homoscedasticity conditions. (1 USD = 9,6 MAD).

SD = standard deviation, Min = Minimum, Max = Maximum.

**Table 5. Descriptive analysis of transport costs for CL-RDT and microscopy.**

| | Diagnostic tool manipulated | N* | Mean | SD | Min | Max | p-value** |
|---|---|---|---|---|---|---|---|
| Minimal Needed visits to PHC and/or Lab to get CL diagnostic | Microscopy | 34 | 2.3 | 0.4 | 2 | 3 | |
| | CL Detect Rapid Test | 6 | 1.0 | 0.0 | 1 | 1 | < .0001 |
| Maximal Needed visits to PHC and/or Lab to get CL diagnostic. | Microscopy | 34 | 3.4 | 0.5 | 2 | 4 | |
| | CL Detect Rapid Test | 6 | 2.3 | 0.5 | 2 | 3 | 0.0008 |
| One-time transport cost from PHC to Lab (Assumption of 4 MAD as a minimum for walking without any public transport use) | Microscopy | 34 | 58.6 | 55.4 | 4 | 180 | |
| | CL Detect Rapid Test | 6 | 58.6 | 73.8 | 4 | 190 | 0 .8 |
| Additional minimal transport cost (assumption PHC is near to each home with no mandatory need to go to Lab) | Microscopy | 34 | 20.8 | 30.4 | 0 | 80 | |
| | CL Detect Rapid Test | 6 | 10.3 | 15.4 | 0 | 40 | 0.7 |
| Additional maximal transport cost (assumption PHC is near to each patient home with the mandatory need to go to Lab) in MAD | Microscopy | 34 | 224.0 | 227.6 | 8 | 720 | |
| | CL Detect Rapid Test | 6 | 124.6 | 145.9 | 8 | 380 | 0.3 |
| Average transport cost in MAD | Microscopy | 34 | 122.4 | 108.7 | 8 | 360 | |
| | CL Detect Rapid Test | 6 | 67.5 | 71.6 | 6 | 190 | 0.1 |

(*) It indicates that among the 40 healthcare professionals participating in the study, six had prior hands-on experience with the CL-RDT. These experienced individuals were from six distinct healthcare facilities, as elaborated in S2 Table. Conversely, the remaining 34 participants were exclusively acquainted with the CL-RDT procedure through a video demonstration.

(**) The p-value was calculated by ANOVA unidirectional for non-parametric tests due to not respecting normality and homoscedasticity conditions. (1 USD = 9,6 MAD).

SD = standard deviation, Min = Minimum, Max = Maximum.

notify a statistically significant preference for CL-RDT over microscopy regarding workload diagnostic cost (p = 0.009) and overall cost at various strip costs (p-values ranging from 0.02 to 0.09). CL-RDT is generally more acceptable to healthcare professionals regarding the time required to perform the diagnostic process and the overall cost, mainly when the unitary strip cost is lower.

**Table 5** reports that patients using CL-RDT typically required fewer healthcare visits, with most needing only a single visit, a significant reduction from the average of 2.3 visits for microscopy. This trend extends to maximum visit counts, transport costs, and additional transport expenses, where CL-RDT consistently demonstrates lower averages and ranges. While the differences in some transport costs are not statistically significant, the overall pattern favours CL-RDT.

**Regression analysis of CL diagnostic costs.** **Table 6** provides a detailed linear regression model, demonstrating the factors influencing the overall CL cost, including workloads, unitary equipment and laboratory material costs.

Table 6 elucidates the various factors affecting the overall cost of CL diagnostics, providing a comprehensive model for understanding cost dynamics. The model's explanatory power is highlighted by its high coefficient of determination ($R^2$ = 0.98), which indicates that the independent variables in the model account for 98% of the variability in the overall CL cost.

Intercept*: The intercept of the regression model, representing the expected value of the overall CL diagnostic cost when all predictor variables are at their reference levels, is 52.3 MAD (Moroccan Dirham).

Diagnostic tool manipulated (Microscopy–CL Detect Rapid Test): Utilising Microscopy instead of CL Detect Rapid Test decreases the cost by 35.8 MAD, and this result is highly

**Table 6. Linear regression predictive model influencing the overall CL cost (workload + the unitary equipment and laboratory material costs).**

| Predictors | Estimation | Standard error | t | *p-value* | Other estimates depending on the unitary strip cost | | |
|---|---|---|---|---|---|---|---|
| | | | | | B | C | D |
| Intercept* | 52.3 | 7.8 | 6.7 | < .001 | 62.0 | 70.5 | 83.1 |
| Diagnostic tool manipulated: | | | | | | | |
| Microscopy / CL Detect Rapid Test | -35.8 | 6.8 | -5.2 | < .001 | -45.4 | -53.8 | -66.4 |
| Previous Knowledge of CL-RDT: | | | | | | | |
| Yes / No | 7.4 | 4.7 | 1.5 | 0.1 | 7.7 | 7.9 | 8.2 |
| HP working time to perform the diagnostic work without delay | 0.4 | 0.02 | 16.1 | < .001 | 0.4 | 0.4 | 0.4 |
| HP needed to perform the full diagnostic test: | | | | | | | |
| Just one / More than one | -3.9 | 4.9 | 0.8 | 0.4 | -3.8 | -3.7 | -3.6 |
| Main Laboratory tech involved in CL diagnostic: | | | | | | | |
| Yes / No | 29.6 | 6.6 | 4.4 | < .001 | 30.4 | 31.0 | 31.7 |
| The patient motif of transfer for the diagnosis process: | | | | | | | |
| Sampling in the PHC without patient transfer / Patient self-laboratory visit | -2.4 | 4.8 | -0.5 | 0.6 | -3.5 | -4.3 | -5.3 |
| MoH Driver involved / Patient self-laboratory visit. | 7.1 | 4.2 | 1.7 | 0.1 | 6.4 | 5.9 | 5.2 |
| Average transport cost | 0.007 | 0.02 | 0.3 | 0.7 | 0.006 | 0.006 | 0.007 |
| Administrative MoH cost (**) | 1.0 | 3.6 | 0.3 | 0.7 | 0.5 | 0.2 | 0.08 |

Note: This model has a high coefficient of determination (R = 0.98), indicating that it explains 98% of the variability in the overall CL cost when the ponderation of RDT strip unitary cost is equal to 45 MAD (1 USD = 9,6 MAD).

(*) is the reference level

(**) free of charge in our context; however, for calculating the coefficient in the model, the cost ranged between 0 and 1 MAD.

B, C, and D present the estimations when the ponderation of RDT strip unitary cost equals 54 MAD, 62 MAD and 74 MAD, respectively.

significant (p < 0.001). Other estimations are 45.4, -53.8, and -66.4 MAD for strip unitary costs of 54, 62, and 74 MAD, respectively. Value: 1 if Microscopy, 0 if CL Detect Rapid Test.

Previous Knowledge of CL-RDT (Yes–No): Prior knowledge of CL-RDT increases the cost by 7.4 MAD, although this effect is not statistically significant at the 0.05 level (p = 0.1). Other estimations: 7.7, 7.8, 8.2 MAD. Value: 1 if yes, 0 if No.

HP working time to perform the diagnostic work without delay: For every additional minute of working time, the cost increases by 0.4 MAD per minute, and this effect is highly significant (p < 0.001). Value Unit: Minutes.

HP needed to perform the full diagnostic test (Just one–More than one): The estimation indicates a decrease in the overall CL cost by 3.9 MAD when just one HP is needed instead of more than one. However, the p-value of 0.4 suggests that this effect is not statistically significant, so the cost reduction might not be a dependable outcome. Value: 1 if just one, 2 if more than one.

Main Laboratory tech involved in CL diagnostic: If the leading laboratory tech is involved, the cost increases by 29.6 MAD, which is highly significant (p < 0.001). Other estimations: 30.4, 31.0, 31.7 MAD. Value: 1 if yes, 0 if No.

The patient motif of transfer for diagnosis process: Effects can vary depending on:

Sampling in the PHC without patient transfer / self-laboratory visits decreases the cost by 2.4 MAD (p = 0.6). Other estimations are -3.5, -4.3, and -5.3 MAD. (Value: 1 if Sampling in PHC, 2 if Sampling & transfer of the sampling, 0 if patient self-laboratory visit)

MoH Driver involvement increases the cost by 7.1 MAD (p = 0.1). Other estimations: 6.4, 5.9, 5.2 MAD. (Value: 1 if MoH Driver involved, 2 if Driver with the patient involved, 0 if patient self-laboratory visit), but, this is not significant (p = 0.5).

Average transport cost: A one MAD increase leads to a 0.007 MAD increase in overall cost (p = 0.7). Value in MAD.

Administrative MoH cost (free of charge): This cost effect is 1.0 MAD and is not statistically significant (p = 0.7). For modelling purposes, the cost ranged between 0 and 1 MAD. Value in MAD.

The linear regression model presented in Table 6 was designed to comprehensively analyse the factors influencing the overall cost of CL diagnostics. Each predictor variable was selected based on its relevance to the cost dynamics of CL diagnosis:

Diagnostic tool manipulated: This variable captures the cost differential between using microscopy and the CL Detect Rapid Test. It is significant in our model, indicating that choosing diagnostic tools substantially impacts cost.

- Previous Knowledge of CL-RDT: Addresses whether prior familiarity with CL-RDT influences the diagnostic cost. Though not statistically significant, it has been included for its potential practical relevance.

- HP working time: Reflects the impact of healthcare professionals' time on the cost, a crucial aspect given the differing time requirements of the diagnostic methods.

- HP needed for the diagnostic test: Explores whether needing more than one healthcare professional affects the cost.

- Main laboratory tech involved: Assesses the cost implications of involving a primary laboratory technician in the diagnostic process.

- Patient motif of transfer for diagnosis process: Includes two scenarios—sampling in the PHC without patient transfer Vs MoH driver involvement—to examine transportation's role in overall costs.

- Average transport cost: Considered to understand how transport expenses contribute to the total diagnostic cost.

- Administrative MoH cost: Although minimal, it has included to ensure a comprehensive cost model.

The strength of Table 6 lies in its high coefficient of determination (R = 0.98), indicating that it explains 98% of the variability in the overall CL cost when the ponderation of RDT strip unitary cost equals 45 MAD. It also reveals how different thresholds of unitary strip cost (54 MAD, 62 MAD, 74 MAD) adapt the initial estimations. While some predictors, such as diagnostic tools manipulated and laboratory technicians involved in CL diagnostic, show consistent effects across different thresholds, others exhibit more minor variations. The table also accounts for the average transport cost. However, this effect is insignificant (p = 0.7), and administrative MoH cost is calculated within a minimal amount due to being free of charge in our context with or without health coverage availability.

Then, the given predictive formula based on the ponderation condition with a unitary cost of 45 MAD for RDT strips is the following:

**Overall CL diagnostic COST** = **52,3**–**35,8**×(Microscopy vs CL Detect Rapid Test) + **7,4**×(Previous Knowledge of CL-RDT: Yes vs No) + **0,4**×(HP working time)– **3,9**× (HP needed to perform the full diagnostic test: just one HP vs more than one) + **29,6**×(Main Laboratory tech involved: Yes vs No)– **2,4**×(Sampling in the PHC without patient transfer vs patient self-laboratory visit) + **7,1**×(MoH Driver involved vs patient self-laboratory visit) + **0,007**×(Average transport cost) - **1,0**×(Administrative MoH cost).

The other predictive formula should use the specific estimations (B, C, D) for each ponderation factor (54, 62 or 74 MAD).

**Sensitivity assessment of CL diagnostic costs across different scenarios.** We conducted a sensitivity analysis to assess the robustness of our cost findings rigorously. This analysis not only underscores the reliability of our results but also sheds light on the relative impact of various factors under different conditions.

The scenarios were meticulously constructed to reflect a range of practical situations in CL diagnostics, providing a comprehensive view of potential cost variations influenced by different clinical and logistical factors.

The results from these scenarios, informed by our sensitivity analysis, illustrate the dynamic nature of CL diagnostic costs, highlighting key variables that exert the most significant influence. The key parameters were:

- Previous knowledge of CL-RDT (binary variable [0, 1]): This parameter considers whether healthcare professionals have prior experience or knowledge of using CL-RDT.

- Diagnostic tool used (Microscopy = 1, CL-RDT = 0, binary variable): Differentiates the costs based on whether microscopy or CL-RDT is used for diagnosis.

- Health professional working time ([20, 62] minutes): Represents the time healthcare professionals spend performing the diagnostic process.

- Number of Health professionals needed (Just one = 1, More than one = 2, binary variable): Assesses the impact of requiring more than one professional for the diagnostic test.

- Patient transport modalities:

○ Self-transfer (Sampling in PHC without patient transfer = 1, binary variable): When patients visit the PHC themselves without needing additional transportation.

○ Driver transfer (MoH Driver involved = 1, binary variable): When transportation involves a driver, possibly increasing costs.

- Patient transport cost ([14 MAD, 432 MAD]): The financial burden of patient transportation.

- Involvement of main laboratory technician (binary variable [0, 1]): Examines the cost effect of involving the main laboratory technician in the diagnostic process.

- Administrative MoH cost ([0 MAD, 1 MAD]): As regulated by the Ministry of Health, administrative costs are minimal in our context.

**Fig 2** displays the total sensitivity indices for crucial parameters influencing the overall CL diagnostic cost at two distinct unitary costs of CL-RDT: 45 MAD and 74 MAD, corresponding to the overall CL-RDT cost of 83 MAD and 112 MAD, respectively. The analysis highlights that the 'Diagnostic Tool Manipulated' parameter substantially impacts 49% to 76% of the overall CL diagnostic cost. Conversely, the 'Main Laboratory Tech Involved' parameter plays a significant but varying role, contributing 34% to 17% of the overall CL diagnostic cost. Other parameters, including 'HP Working Time,' exhibit more modest influences. At the same time, factors like 'Previous Knowledge of CL-RDT,' 'HP Needed,' 'Self Transfer,' 'Driver Transfer,' 'Patient Transport Cost,' and 'Administrative MoH Cost' contribute marginally.

Python analysis defined the CL-RDT unitary cost range compared to microscopy cost (115 MAD). As explained in the method section, we employed the SALIb library with the Sobol method to quantitatively assess the sensitivity of our cost model to changes in the various input parameters, ensuring a thorough and robust analysis (**S1 Codes**).

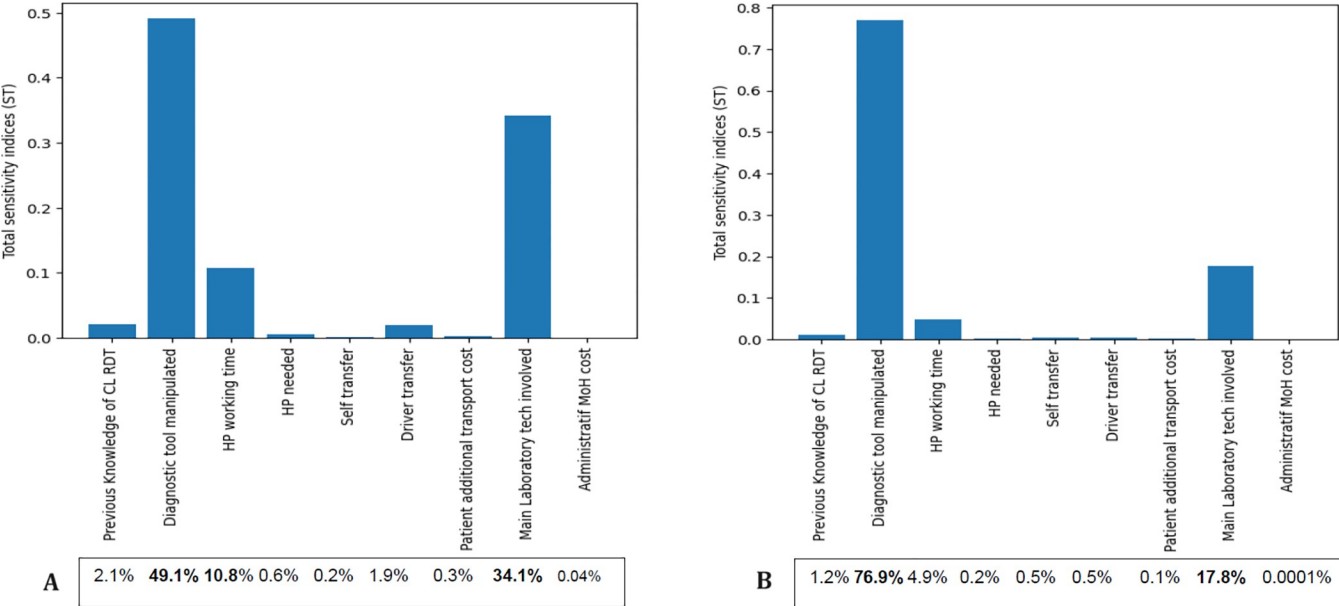

**Fig 2.** A & B: Comparison of total sensitivity indices for key parameters influencing overall CL diagnostic cost at two unitary costs of CL-RDT—45 MAD (2A Left) and 74 MAD (2B Right). **Bounds conditions**: # previous_knowledge (binary variable) [0, 1]; # diagnostic_tool (Microscopy = yes = 1) (binary variable) [0, 1]; # hp_working_time [20, 62]; # hp_needed (just one = 1) (binary variable) [1, 2]; # self_transfer (Sampling in the PHC without Patient transfer = 1) [0, 1]; # driver_transfer (MoH Driver involved = 1) [0, 1]; # patient_transport_cost [14 MAD, 432 MAD]; # main_lab_tech_involved (binary variable) [0 1]; # admin_moh_cost [0 MAD, 1MAD].

Then, real-world scenarios were simulated to examine how parameter changes could affect the cost of diagnosing cutaneous leishmaniasis, thereby calculating cost-benefit rates.

To summarise, **Fig 3** shows five CL diagnostic scenarios. **Scenario 1,** where CL-RDT is available at the PHC, the diagnostic will be completed in under 30 minutes by only one HP who will provide the result to the patient on the same day, without needing laboratory equipment or technician expertise, nor additional transport cost nor additional administrative cost. The CL-RDT unitary cost is more advantageous than microscopy if the cost is lower than **83 MAD**.

However, in **scenario 2**, CL-RDT is available at the laboratory and used by lab technicians within the same conditions as in scenario 1. The minimal timing remains the same. Then, the CL-RDT unitary cost is advantageous if the cost is lower than **54 MAD**.

**In scenario three,** where microscopy should confirm the initial result of CL-RDT, the minimal time needed will be 75 minutes with the need of at least 2 HP with the involvement of the laboratory technician. Then, CL-RDT unitary cost remains advantageous if the cost is lower than **40 MAD**.

**In scenario 6, the CL-RDT is provided twice at the PHC by the same HP for confirmation diagnostic without** microscopy. The 2nd CL-RDT unitary cost assumption was defined as 62 MAD. Then, CL-RDT unitary cost remains advantageous if the cost is lower than **09 MAD**. The other scenarios and the main variation of cost thresholds are shown in **Fig 3**. Furthermore, almost 49% to 75% of the overall CL diagnostic cost is linked to the unitary cost of the strip.

As reported in **Fig 4**, the normalised Cost-Benefit Ratio (CBR) was calculated for all proposed scenarios compared to scenario three as a reference (Doing CL-RDT at PHC and then visiting the laboratory for microscopy). CL-RDT done only at the PHC is the best scenario (CBR scenario1 = 1,4) while doing only CL-RDT at the laboratory is the second-best scenario (CBR scenario2 = 1,0). Facing additional transfer and/or transport costs and/or administrative

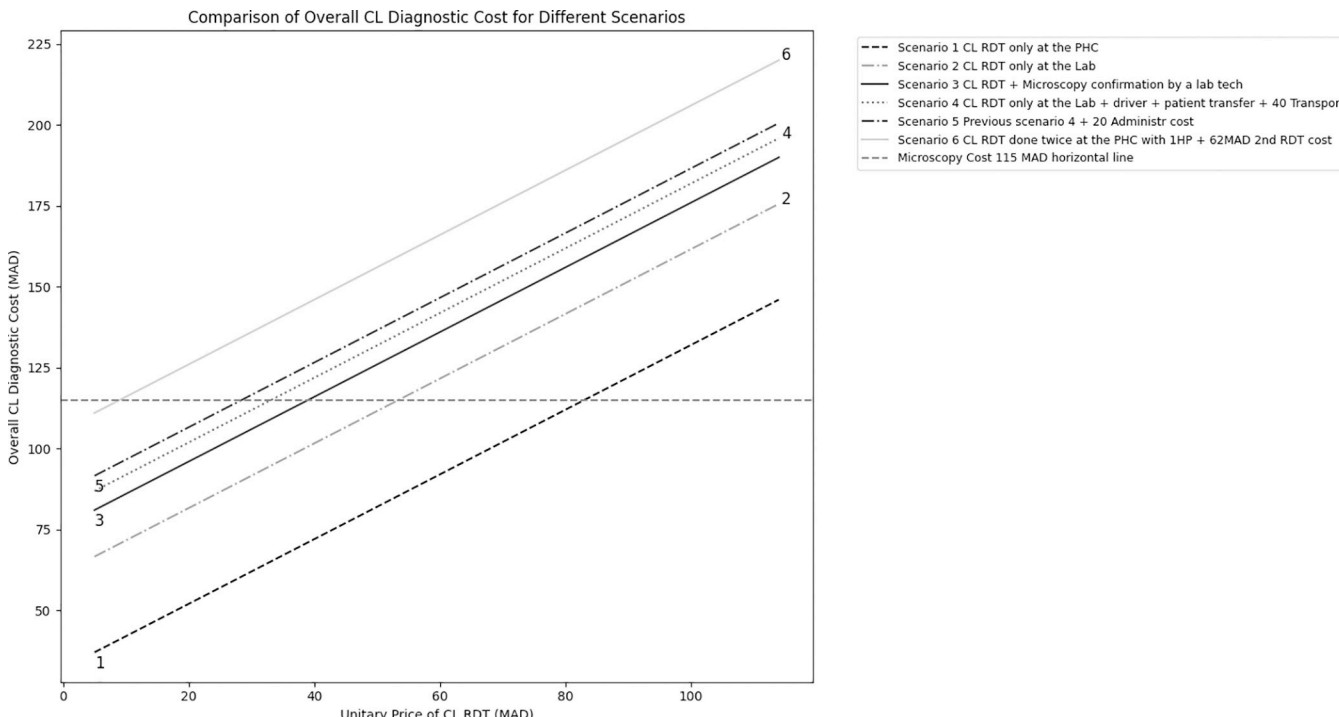

**Fig 3. Comparison of overall CL rapid diagnostic costs across microscopy cost threshold (115 MAD).**

costs in scenarios 4 and 5 decreases the benefit (CBR < 0,9). In addition, for scenario one, the CL-RDT unitary cost range remains advantageous between 25 and 79 MAD. In contrast, the range becomes smaller between 25 MAD and 54 MAD in scenario 2.

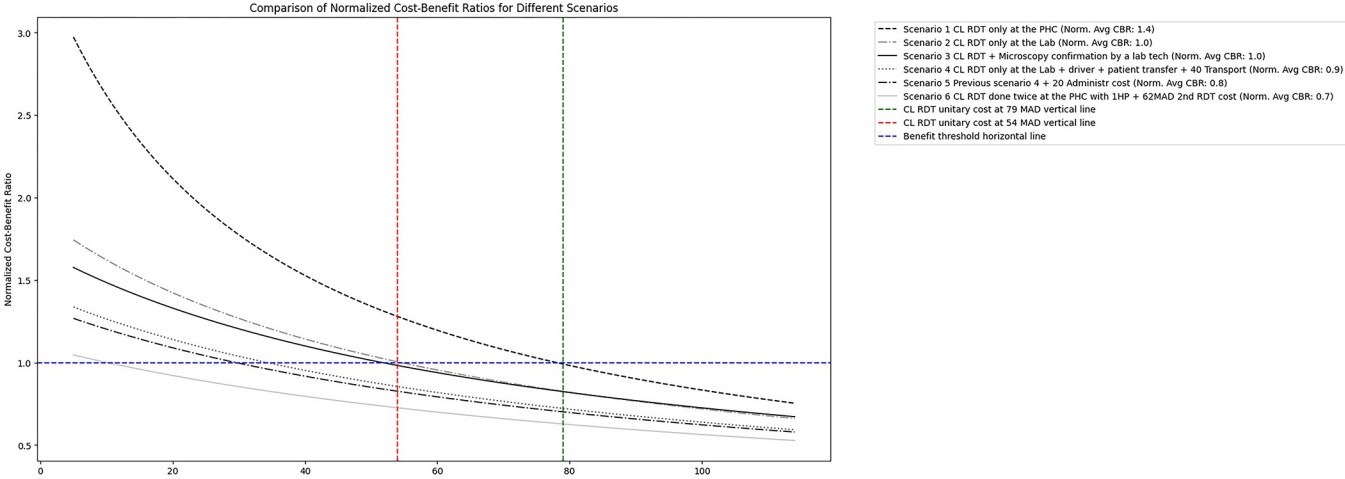

**Fig 4. Normalised cost-benefit ratios for different scenarios compared to scenario 3.** Assumptions were made for calculating the benefit value based on the test accuracy, the test acceptability, the saving of patient time and the laboratory reliability. # Test accuracy calculation (= True positive rate + true negative rate) TPR = Sensitivity * Prevalence TNR = Specificity * (1-Prevalence) # CL-RDT Se 68% Sp 94% PPV 95% NPV 64% (Bennis et al 2018 [9]) # Microscopy Se 65% Sp 100% PPV 100% NPV 16% (Bennis et al 2018 [9]) # Prevalence was considered equal to 50%, and laboratory reliability was considered the same for all scenarios at 100%, # Scenario 3 is the reference scenario for calculating the CBR. While Scenario 3 includes microscopy cost as115 MAD threshold #accuracy # acceptability # saving_patient_time # laboratory_realibility #1 CL-RDT only at the PHC 0.81 1 0.9 1 #2 CL-RDT only at the Lab 0.855 0.5 0.7 1 #3 CL-RDT + Microscopy confirmation by a lab tech 0.94 0.7 0.5 1 #4 CL-RDT only at the Lab + driver + patient transfer + 40 Transport 0.855 0.3 0.2 1 #5 Previous scenario 4 + 20 Administr cost 0.855 0.1 0.1 1 #6 CL-RDT done twice at the PHC with 1HP + 62 MAD 2nd RDT cost 0.855 0.9 0.7 1.

**Table 7. Summary of the health professional's in-depth interviews' analysis of cutaneous leishmaniasis diagnostics.**

| Main Theme | Sub-Theme | Description |
|---|---|---|
| **Diagnostic process and efficiency** | **Workload comparison** | The workload for CL-RDT is lower than for microscopy.<br>Healthcare professionals prefer CL-RDT to improve diagnostics and increase patient compliance.<br>Concerns about increased workload in rural PHCs due to healthcare professional shortages. |
| | **Usefulness in different areas** | The perceived usefulness of CL-RDT varies: it is more appreciated in *L. tropica* areas (Tinghir, Sefrou provinces) compared to *L. major* areas (Errachidia, Ouarzazate). |
| | **Training and familiarity with CL-RDT** | Laboratory technicians find the procedure straightforward, but not doctors and nurses.<br>Familiarity with CL-RDT among healthcare professionals leads to ease and proficiency in use. |
| | **Practicality of CL-RDTs** | Mixed opinions on the practicality of CL-RDTs in primary health centres.<br>Some find it easy and beneficial, reducing transportation costs for patients.<br>Others rely on clinical decisions and epidemiological trends.<br>Training and consistent testing across health centres and laboratories are suggested. |
| | **Comparative analysis with microscopy** | CL-RDT does not require formal training and can be performed by any healthcare professional, potentially testing more patients per day than microscopy.<br>Microscopy requires at least two weeks of training and can be emotionally stressful. |
| **Expected patients' preferences** | **Acceptability and comfort** | Acceptability of CL-RDT varies based on healthcare professionals' familiarity with it.<br>Concerns about discomfort from the dental broach are alleviated by local anaesthetic.<br>No reported pain or damage from dental broach use. |
| | **Waiting time for results** | The acceptable waiting time for microscopy results is 5 hours, but the actual wait can be 2 to 4 working days, sometimes over 15 days.<br>Delays are due to technical issues, lack of training, and scheduled weekly activity organisation.<br>Patients are unwilling to wait for microscopy results, preferring quicker CL-RDT results. |
| **Economic aspects and healthcare policy considerations** | **Resource allocation and training** | Insufficient trained healthcare professionals or laboratory technicians could result in service interruptions.<br>Suggestions for improved motivation mechanisms from the Ministry of Health and training one healthcare professional and one lab technician in each endemic area. |
| | **Use in remote areas and by non-health workers.** | Non-health workers performing RDTs in remote areas are not accepted due to concerns about storage conditions, ambient temperature, expiration dates, and regulations.<br>Pharmacists and private drugstores could conduct CL-RDTs, provided tests are reimbursed by the health coverage system. |
| | **Patient's willingness to pay** | Patients prefer undergoing the diagnostic process at the same PHC to avoid extra costs like transportation.<br>The suggested cost for CL-RDTs in drugstores should not exceed 50 MAD.<br>Patients consent to treatment when positive test results indicate increased compliance with CL-RDT. |

## The qualitative results linked to CL diagnostic

Initial interviews were conducted by the principal investigator's team, followed by six focused interviews at the end. No differences in responses based on gender were found. However, differences were observed in other attributes, as explained below and summarised in **Table 7**.

The quantitative information from the first section of the results was qualitatively confirmed (**S6 Table**). The workload needed for CL-RDT is lower than that for microscopy. From the healthcare professional's perspective, many patients prefer undergoing the diagnostic process at the same PHC to avoid extra costs such as transportation expenses and time delays.

*"The result of the microscopy is available in one day, a maximum of 48 hours, but the problem is the delayed patient because they do not have the means to follow up on their health condition quickly. They generally return home and try to save money for these expenses. Then, for example, a woman is obliged to go out only once a week, which is already dictated by her*

*husband, and her husband must give her permission to accompany her if he is not with her. Moreover, the schedule is not suitable because the people are far away, and even if they come, they will arrive after 2 PM; for example, there is a person who comes on foot for a kilometre, and when he comes to the dispensary, you ask him to go another 40 kilometres to the nearest town, so he is not going to do that."* Nurse L.tropica area.

Participants believed the acceptable waiting time for microscopy results should be 5 hours [2h-10h]. However, patients receive their results between two to four working days, with some cases taking over 15 days. Delays are attributed to technical problems, the lack of training for nurses and doctors in performing skin smears, the absence of known standards for methanol and Giemsa staining by novice lab technicians and the scheduled weekly activity organisation.

*"Because it is not an urgent examination, the result is not urgent, so we can afford to wait a day or two to read the slides and make a high-quality diagnosis properly."* Laboratory technician L. major area.

They view CL-RDT as a solution to improve CL diagnostics and increase patient compliance. However, there is a concern about the increased workload due to the new diagnostic tool in rural PHCs due to the HP shortage.

The perceived usefulness of the CL-RDT test compared to microscopy varied between health workers in different CL areas. It was highly appreciated in *L. tropica* areas (Tinghir, Sefrou provinces) but less so in *L. major* areas (Errachidia, Ouarzazate) due to the easy clinical diagnosis for typical forms of *L. major* CL cases.

*"No, never (asking microscopy confirmation); I treat all cases based on symptomatology because here, the population refuses to travel to the main town to do the analysis."* Physician L. major area.

*"For the diagnosis of CL, it is normally difficult; there are many similarities between the lesions, so we should not rely solely on the clinical examination."* Nurse L.tropica area.

*"Here, in an area where there is Leishmania tropica and Leishmania infantum, we are supposed to perform a laboratory examination to confirm the diagnosis."* Physician L.tropica area.

*"Each time there is an epidemic peak, hundreds of people are affected, so there is no need to send them to the laboratory to confirm the diagnosis; we treat them here without going back and forth."* Nurse L. major area.

The qualitative analysis revealed acceptability differences based on participants' previous use or knowledge of CL-RDT:

Acceptability of healthcare professionals who were unfamiliar with CL-RDT before the study (N = 23; Men = 9):

Healthcare professionals who had never used CL-RDT expressed concerns about potential patient discomfort, which could be alleviated with local anaesthetic ointments or injections. Despite this being an exclusion criterion, they were apprehensive about using a dental broach near patients' eyes.

The video demonstrating the CL-RDT procedure, which involves a superficial double rotation of the dental broach, was unfamiliar to them. Some professionals noted that they sometimes could not find parasites in typical CL lesions even after taking multiple scraping samples, suggesting that the small broach and twisting technique may be insufficient.

The CL-RDT procedure using lysis buffer, Eppendorf tubes, tips, pipettors, and chase buffers was familiar and straightforward for laboratory technicians but not doctors and nurses, as it was not part of their previous training. One doctor cited safety concerns when transferring mixtures from the tube to the RDT strip as a problem (The manufacturer recommends using gloves and protective eyewear to minimise risks).

Additionally, they worried about finding a quiet space in the PHC to perform the RDT without interruptions from other patients. The limited number of healthcare professionals working in the same PHCs to help manage this new activity within their daily routines underscored the need for improved motivation mechanisms from the Ministry of Health. Moreover, insufficient trained healthcare professionals or laboratory technicians involved in CL management could result in service interruptions during annual holidays or exceptional leaves.

*"Being an urban health centre and as the doctor of this centre, I prefer to send patients directly to the nearby laboratory in the same city to carry out the smear, the sampling, and the analysis of the microscopic reading result."* Physician L. major area.

*"For the staff, there is an overload for all the activities performed, and we will ask them to add the handling of these rapid tests as well. Due to the lack of personnel, this will be a disadvantage."* Nurse L. major area.

<u>Healthcare professionals who were familiar with or had previously used CL-RDT (N = 23; Men = 17)</u>

They appreciated CL-RDT in actual practice for both providers and patients. They found that the video demonstration made it easy to understand the new sampling technique without requiring additional training. One participant suggested adding information about handling errors and invalid strip results. As they gained experience with CL-RDT, healthcare professionals quickly became confident and proficient, even remembering how to perform it by watching the video demonstration.

*"The video shows that performing the rapid test for CL is simple and not complicated. It's an easy test that can be conducted even in times of heavy workload because there weren't too many leishmaniasis cases last year."* Nurse L. major area.

In contrast, learning microscopy requires at least two weeks of training at a reference laboratory, provided the microscopist has good vision and concentration. New digital microscopes that can record videos and images for display on larger screens can assist laboratory technicians in making observations more easily.

*"I am old, and I have a vision problem, and I cannot see the result easily, so I prefer to take the sample and send it for microscopy rather than perform the rapid test. But if we have someone younger to perform this rapid test, we can easily use it at the health centre."* Nurse L. major area.

Interviewees confirmed that a technician can analyse approximately 4 to 20 slides per day using microscopy (with at least two smears per patient), which can be repetitive and lead to emotional stress or burnout. CL-RDT, on the other hand, does not require formal face-to-face training and can be performed by any healthcare professional, with the potential to complete 10 to 30 tests per day. That could allow up to 5 times more patients to be tested per day per healthcare professional compared to microscopy.

*"The rapid test takes 30 minutes, and microscopy takes about 1 hour to make a smear for a single lesion. It will take you 30 minutes for the microscopy and then about 45 more minutes because, in the end, you take the sample, spread it on the slides, and let it dry. Then, you fix it for a minimum of 5 minutes, then apply the stain, which takes about 45 to 50 minutes. Then you wash it and let it dry again before adding a drop of oil to start reading field by field. It takes more than an hour unless you do a sloppy job or immediately find the amastigote bodies of leishmaniasis. If you are lucky and not disturbed by other tasks, it varies between one hour fifteen minutes to one hour and half. So, for me, the rapid test is better for me and also better for the patients, and I would like to use it directly in the laboratory." Laboratory technician L. tropica area.*

All prior HP users of CL-RDT mentioned that patients consented to treatment when the test results were positive. Additionally, participants explained that patients were unwilling to wait for microscopy results and some experienced minor injuries from the scraping technique for smear slides. However, no patients reported pain or damage from the dental broach, even those who opted not to use local anaesthesia (Lidocaine).

*"There are people who will decide to go to the laboratory while others will not go because they have a problem with availability of time; they have a job that they cannot abandon. . . If the patient has to travel, they will lose time, and they will lose even more time if they do it just here on site, especially when they have to travel; it's not certain if the other laboratory will be available or not, whether the technician is present or not, whether they will do the reading on the same day or not." Nurse L.tropica area.*

In-depth exploration of the acceptability and practicality of CL-RDTs

Throughout the interviews, opinions varied on enhancing CL diagnostics in primary health centres and who should perform the tests if they become widespread. Some professionals found it easy enough to act alone and beneficial as a point-of-care tool, mainly that would reduce patients' transportation costs. They noted that RDT results are available within 40 minutes. However, others argued that they might not have the time, especially during an outbreak, and would instead rely on clinical decisions and epidemiological trends, as the Ministry of Health recommended for *L.major* areas.

Some interviewees stated that nurses should not make treatment decisions based on the RDT results alone and should rely on external lab tests or medical approval. For example, one nurse shared that they had received RDTs for malaria and tuberculosis but did not use them for similar reasons. In contrast, others believed that using CL-RDTs differed since healthcare professionals were accustomed to initiating standard treatments without laboratory confirmation during outbreaks.

*"If there is management through the rapid diagnostic test, the lesions will be less complicated. Therefore, the presence of rapid tests will even improve the treatment outcome, and people will come more often to receive care at health centres and dispensaries. There will be less abandonment of treatment." Nurse L.tropica area.*

Another doctor suggested training one healthcare professional and one lab technician in each known or new endemic area to ensure consistent testing across primary health centres and laboratories. However, some participants believed that CL-RDTs should be performed exclusively at PHCs, with microscopy used for confirmation in case of doubtful results. Others preferred performing a second RDT targeting a different part of the lesion or a new one

instead of waiting for microscopy results, which was widely agreed upon if the first RDT was negative.

> *"Now that the rapid test is not yet widespread, people (Health professionals) will not understand its usefulness or benefits. But when it becomes available and widespread, and the first people start to use it, they will realise the difference, and in that case, they will only accept this rapid test." Nurse L.tropica area.*

The idea of allowing non-health workers to perform RDTs in remote areas under healthcare professionals' supervision was not accepted due to concerns about storage conditions, ambient temperature, expiration dates, and nursing practice law regulation. Nevertheless, many participants recognised the potential role of pharmacists and private drugstores as suitable locations for conducting CL-RDTs, provided that the Moroccan health coverage system could reimburse such tests. Based on participants' experiences, the suggested cost for CL-RDTs, if made available in drugstores, should not exceed 50 MAD [20–200].

> *"If it can be made available for free by the Ministry of Health at health centres, then that would be ideal." Physician L. major area.*

> *"For me, a cost of 30 MAD would be acceptable, and it should not exceed 50 MAD at all. The person in charge at the pharmacy must be competent.". Nurse L. major area.*

## Discussion

Most HPs appreciated the convenience of the point-of-care CL-RDT at PHCs. CL-RDT's primary advantage is that it can be performed by a single HP for patients with suggestive CL lesions, providing rapid results during the patient's initial visit to the PHC without additional transportation costs or time delay. For example, in outbreaks or emergencies (e.g., COVID-19), CL-RDT limits transportation requirements and minimises the number of patient interactions for HPs.

Furthermore, participants generally believed that patients with an initially negative CL-RDT test would prefer a second RDT over waiting for microscopy results, highlighting the broad acceptability of this diagnostic tool to be provided at the first contact point with the patients in the Moroccan context. Indeed, the Cost Benefit Ratios for a scenario where CL-RDT is only done as the primary CL diagnostic at the PHC or where CL-RDT is done alone at the laboratory were both more advantageous than the reference scenario where microscopy is done systematically at the laboratory if the CL-RDT done at the PHC was negative.

However, doing a CL-RDT twice at the PHC to confirm the negative initial diagnostic was not an advantageous scenario (CBR = 0,6) except if the CL-RDT unitary cost will be purchased lower than 09 MAD or making the decision to support the 36 MAD overall cost difference between this scenario and the reference one (assuming CL-RDT unitary cost equal to 62 MAD).

The overall CL diagnostic costs are impacted by the strategies defined for CL control (which could demand targeting the best diagnostic confirmation scenarios) and the general acceptability of the diagnostic tool by healthcare professionals (which can influence how often the tool is used or manipulated).

Our findings align with very few studies that targeted CL diagnostic costs rather than determining only the test's accuracy. Indeed, the studies from Peru and Sri Lanka agreed on the need for low-cost CL diagnostic tests, defined (as 4–5 USD per test) [12,25].

However, our study extends these cost insights by quantifying the micro-costing advantage of CL-RDT and exploring its acceptability in the Moroccan context. Unlike another study from Columbia, which focused on the cost analysis of PCR kDNA test for mucocutaneous leishmaniasis diagnostic, that could be recommended for future research for localised cutaneous leishmaniasis areas [26].

Our study highlighted at least the same costs of CL RDT compared to microscopy. This is almost the same finding in a previous study where RDT was comparably cost-effective to microscopy at reference clinics (53 USD) but offered ease of use, making it cost-effective at peripheral levels (48 USD) with no differences in costs between the RDT and microscopy [22].

Additionally, pairing the CL Detect Rapid Test with microscopy for negative cases at the reference clinic was more economical than using microscopy alone for all CL suspects [22]. In comparison, a recent study emphasises the usefulness of CL-RDT in case of limited access to expert microscopists [11]. The interest increases in being detected as a travel medicine disease in non-endemic CL countries where the average diagnostic delay could be more than 180 days [27].

Some explanations for this study's findings open areas of reflection:

Firstly, the WHO has historically recommended making Rapid Diagnostic Tests (RDTs) accessible at the most readily available point-of-care facilities [28]. Furthermore, the expert panel has agreed to develop affordable rapid diagnostic tests for dermal leishmaniasis tailored explicitly for low and middle-income countries [1]. Nevertheless, recent guidance from the WHO Diagnostic Technical Advisory Group for neglected tropical diseases (NTDs) highlights the need to develop new point-of-care tests for Cutaneous Leishmaniasis (CL). The currently available CL Detect Rapid Test fails to meet the requisite diagnostic performance standards of reaching 95% sensitivity for point-of-care waiting for the next generation of CL-RDT based on antigen detection that should differentiate between *L.tropica*, *L.major*, and other species [1,29,30].

Certainly, real-time PCR offers the best accuracy compared to microscopy for any sampling technique [31,32], especially from scraping smears [33] and could be defined as the reference standard test for CL diagnosis [34]. However, it is not yet easy to perform compared to RDT. Other diagnostic techniques like Loop-Mediated Isothermal Amplification (LAMP) and nested PCR are also in development [35,36].

The high acceptability of CL-RDT in *L. tropica* areas compared to *L.major* regions is not solely based on the expected sensitivity and specificity of this test about microscopy. Instead, it is also influenced by the overall time management of CL patients. In *L.major* areas, during new outbreaks, the Moroccan MoH CL management program recommends microscopy only for the first two CL cases in the same locality to confirm the diagnosis. Then, it begins treatment based on clinical presumptions for other new patients from the same area. In *L. tropica* areas, however, microscopy is recommended for each suspected CL patient.

Consequently, the systematic use of a diagnostic test in *L.major* regions could be perceived as an unwelcome additional workload, potentially contributing to HP burnout and negative patient-health provider interactions [37,38].

The psychosocial burden prevalent in some areas, where there is an urgent need to minimise scarring through prompt treatment, underscores the importance of using diagnostic tools like the current CL Detect Rapid Test (CL RDT). With its high specificity, as indicated by the review's pooled analysis showing an overall specificity of 94% [87–97%], the CL RDT or similar rapid diagnostic tools can provide quick answers, essential for ruling out CL [30]. This is especially relevant in Morocco, reflecting a pattern observed in Sri Lanka, where, for the *L. donovani* species, the RDT demonstrated a sensitivity of 36% but a specificity of 100% [25]. A recent study from Sri Lanka highlights the significant public health concern posed by the

psychosocial impact of leishmaniasis, necessitating a holistic approach that addresses the physical, psychological, and social aspects of affected CL patients [39].

Concurrently, comprehensive acceptability studies and cost-advantage analyses are needed to evaluate the potential for their widespread adoption and utility. For example, using drugstores as sites for administering the RDT could help maintain its acceptability in areas lacking qualified HPs.

The perception of improving the quality of care for CL management in primary health centres supports HPs' acceptability of the test and perceptions of its advantages, as seen with malaria RDTs in Ghana and Mali [16,40].

This study has limitations. As a pilot study, it could not target the maximum variation of more involved HPs. However, we enrolled all HPs in Morocco who had previously performed CL-RDT and a homogeneous group that had never heard of it, helping us reach saturation from the HPs' perspective. The study areas were endemic to CL, and the ulcerative form is Morocco's most common clinical manifestation (75 to 85%). Consequently, the results of this RDT for nodular or popular forms, atypical CL cases, or sporadic cases remain unanswered.

The potential challenges of implementing and managing multiple types of RDTs in the same healthcare setting were not explored. With few RDTs currently available in Morocco at the same PHCs, it is challenging to determine professionals' thoughts and expectations regarding the workload of managing various RDTs targeting different diseases and population groups in the same space and time.

The diagnostic performance of the RDT in comparison to microscopy was not thoroughly discussed with healthcare professionals, which may have influenced their perception and acceptability of the test. Either the study did not account for the potential variability in the sensitivity of different sampling techniques depending on lesion type and duration. In contrast, a sensitivity analysis was conducted to examine these factors to overcome the absence of specific details regarding the range assumptions of full transport and administrative costs.

Lastly, the patient's perspective was not directly targeted, which could provide valuable insights into their decision-making and potential cultural and financial barriers to accessing CL healthcare services, such as indirect non-medical opportunity costs. Even if healthcare professionals understand the patients' occupations and their behaviours for obtaining CL diagnostic, in the study areas, these costs might be minimal, as housework, farming, and agricultural activities are the main occupations and can be shared with family members or neighbours, allowing for temporary absences without impacting daily earnings. Conversely, those who risk losing a day's work might avoid visiting primary healthcare centres (PHCs) and rely on traditional medications, as reported in previous studies [3,41–43].

It is imperative to recognise the intricate cost structure categorised into fixed, variable, and combined costs as conceptualised in our context (**Fig 1**), each playing a pivotal role in the overall financial framework. Notably, the green indicators in our analysis emphasise the dynamic nature of these costs, influenced by factors such as healthcare professionals' acceptability of diagnostic tools, policy-driven CL control strategies, and the specific diagnostic confirmation scenarios employed. The general acceptability of diagnostic tools among healthcare professionals directly affects their usage frequency, thereby influencing variable costs. Strategically defined control policies can optimise the diagnostic workflow, resulting in more efficient use of resources and reduced costs. Furthermore, the selection of diagnostic confirmation scenarios determines the allocation of both fixed and variable resources, underpinning the importance of strategic decision-making in the economic management of CL diagnostics.

These findings advocate for policy shifts towards adopting CL-RDT in endemic areas in the Moroccan context. Future research should focus on expanding the accessibility of CL-RDT and exploring its application in diverse epidemiological settings. Depending on the CL species

and clinical presentation, the cost-effectiveness of CL-RDT should be compared to microscopy for multiple lesions. Secondly, both microscopy and CL-RDT should improve their sampling procedures to obtain higher parasite loads with fewer invasive, repetitive lesions, increasing confidence in both tests. The sensitivity of CL-RDT was low for *L.ethiopica* in Ethiopia and *L. donovani* in Sri Lanka [25,44]. The sampling site of lesions could influence parasitological diagnosis sensitivity [45]. Therefore, the development of improved testing methods is crucial.

## Conclusion

Our study indicates that Rapid Diagnostic Tests (CL-RDT) have a micro-costing advantage over microscopy in diagnosing cutaneous leishmaniasis. The average cost of CL-RDT is 100 MAD (ranging from 79 to 131 MAD), compared to microscopy, which averages 115 MAD (ranging from 58 to 292 MAD). This cost includes workload and transport expenses, where CL-RDT demonstrates significantly lower costs, particularly in transport.

Moreover, healthcare professionals prefer CL-RDT due to its ease of use and quick results. In rural areas endemic to CL, the time efficiency and micro-costing advantage of CL-RDT make it a precious tool for CL control management. The most significant micro-costing benefit is observed when CL-RDT is used as the primary healthcare centre's primary diagnostic method, especially if the RDT strip cost remains below 79 MAD.

In anticipation of forthcoming versions with improved sensitivity, deploying the next-generation CL-RDT in resource-limited settings is advocated.

## Supporting information

**S1 Video. CL-RDT video demonstration.**
(DOCX)

**S1 Fig. Zoonotic and anthroponotic cutaneous leishmaniasis cases from 1998 until 2020 in Morocco.**
(DOCX)

**S1 Text. Topic guide interview for HPs.**
(DOCX)

**S2 Text. Thematic guide for the in-depth personal interview.**
(DOCX)

**S3 Text. Questionnaire of laboratory and primary health centres' professionals about CL diagnostic (RDT, microscopy).**
(DOCX)

**S1 Table. Cutaneous leishmaniasis 2019's epidemiological data and hypothetic yearly CL lesions in the study area.**
(DOCX)

**S2 Table. List of anonymised participants, their localisation, profile, and previous knowledge about CL-RDT.**
(DOCX)

**S3 Table. The unitary equipment and laboratory material costs for both CL microscopy and RDT expressed in MAD (1 USD = 9,6 MAD).**
(DOCX)

**S4 Table. The Standards for Reporting Qualitative Research (SRQR) checklist.**
(DOCX)

**S5 Table. The consolidated Health Economic Evaluation Reporting Standards (CHEERS 2022) checklist.**
(DOCX)

**S6 Table. Extraction of the full qualitative tree codes by NVivo software analysis.**
(DOCX)

**S1 Codes. Python scripts for global sensitivity analysis and cost-benefit analysis in cutaneous leishmaniasis diagnostic tools management.**
(DOCX)

## Acknowledgments

We acknowledge the late Pr Marleen Boelaert, who initiated this study but unfortunately passed away in June 2020. She was a professor and head of the Unit of Epidemiology and Control of Tropical Diseases at the Department of Public Health at the Institute of Tropical Medicine in Antwerp. This work was supported at the protocol and initial steps by Pr Albert Picado, and we extend our acknowledgements to Pr Vincent De Brouwere for his orienting advice and support. Our declarations also go to Mr Smain Chichaoui, Dr Abdelatif Hamdaoui, and Mr. Khalid El Houma from the Regional Directorates of the Ministry of Health in the Deraa Tafilalet Region and Fez Meknes Region. The authors acknowledge the technical support of the Moroccan Ministry of Health, represented by the Directorate of Epidemiology and Diseases Control in Rabat and the National School of Public Health in Rabat.

## Author Contributions

**Conceptualization:** Issam Bennis, Naoual Laaroussi.

**Data curation:** Issam Bennis, Abdelkacem Ezzahidi.

**Formal analysis:** Issam Bennis.

**Funding acquisition:** Issam Bennis.

**Investigation:** Issam Bennis, Mohamed Sadiki, Abdelkacem Ezzahidi.

**Methodology:** Issam Bennis, Mohamed Sadiki, Naoual Laaroussi, Souad Bouhout.

**Project administration:** Issam Bennis, Mohamed Sadiki, Abdelkacem Ezzahidi, Souad Bouhout.

**Resources:** Issam Bennis, Naoual Laaroussi.

**Software:** Issam Bennis.

**Supervision:** Issam Bennis, Mohamed Sadiki, Abdelkacem Ezzahidi, Souad Bouhout.

**Validation:** Issam Bennis, Abdelkacem Ezzahidi, Souad Bouhout.

**Visualization:** Issam Bennis.

**Writing – original draft:** Issam Bennis.

**Writing – review & editing:** Issam Bennis, Souad Bouhout.

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
