## [Decision Letter · Decision Letter 0]

13 Dec 2023

PGPH-D-23-01769

Micro-costing from healthcare professional's perspective and acceptability of cutaneous leishmaniasis diagnostic tools in Morocco: A Mixed-methods study

Dear Dr. Bennis,

Thank you for submitting your manuscript to PLOS Global Public Health. After careful consideration, we feel that it has merit but does not fully meet PLOS Global Public Health’s publication criteria as it currently stands. Therefore, we invite you to submit a revised version of the manuscript that addresses the points raised during the review process.

We look forward to receiving your revised manuscript.

Kind regards,

Andrés F. Henao-Martínez, M.D.

Academic Editor

Journal Requirements:

Additional Editor Comments (if provided):

Reviewers' comments:

Reviewer's Responses to Questions

**Comments to the Author**

1. Does this manuscript meet PLOS Global Public Health’s publication criteria? Is the manuscript technically sound, and do the data support the conclusions? The manuscript must describe methodologically and ethically rigorous research with conclusions that are appropriately drawn based on the data presented.

Reviewer #1: Yes

Reviewer #2: Partly

2. Has the statistical analysis been performed appropriately and rigorously?

Reviewer #1: No

Reviewer #2: No

3. Have the authors made all data underlying the findings in their manuscript fully available (please refer to the Data Availability Statement at the start of the manuscript PDF file)?

Reviewer #1: Yes

Reviewer #2: No

4. Is the manuscript presented in an intelligible fashion and written in standard English?

Reviewer #1: Yes

Reviewer #2: No

5. Review Comments to the Author

Reviewer #1: Congratulations to the authors for their work. Please see below comments for consideration

1.Cross check the entire document for correct grammar and punctuation eg " In North Africa, Cutaneous Leishmaniasis" reveals inappropriate use of capital letters

2.Authors are advised to justify their selection of the intervention test.

3.I recommend statements that are not author's own should be supported by references. The paragraph below for example hardly has references

" In Eastern Mediterranean countries where Cutaneous Leishmaniasis is endemic, the standard

laboratory test involves light microscopy examination of a smear prepared by scraping a

suspected lesion. This method enables the visual identification of parasites following Giemsa

stain colouration. Ideally, results are obtained in approximately one and a half hours when

colouration and microscopy readings are performed directly. However, in many remote areas

where people rely on primary health centres (PHCs), patients often require two to three PHCs

and lab visits over several days to receive microscopy results."

4.What is the burden of leishmaniasis in the selected areas of study?

5.It is not clear why the sampling process led to a predominantly high number of nurses? Are they the ones who mainly offer clinical care to the patients? Perhaps a brief overview of health service proivision in Morocco will help put the findings in context

6.How exactly were the interviews conducted? Did all the 40 participate in an interview all at once? How was consenting done and confidentiality maintained?

7.I propose supporting quotes be added to the text to support the deductions made from the qualitative arm of the research and summary table of the qualitative findings highlighting the themes and sub-themes be added to the text

8.I propose that in the discussion section, the authors compare the similarities and differences in their studies with other similar studies and conclude by giving their thoughts on new directions in advocacy, research and clinical practice

Reviewer #2: Thank you for the opportunity to review this manuscript. The manuscript should be re-organized to achieve publication quality. Please find my comments and suggestions below:

* Provide more details motivating the study, i.e. what is the CL prevalence, distribution of disease (age, sex, etc.), what are the estimated coverage rates for CL diagnostic tests, what is the national management protocol for CL

* Methods: it should be clearly stated that the analysis unit is the HP practice (n=46)

* Remove unnecessary details (order of authors when describing responsibilities, credentials of individuals capturing data in Excel)

* The results section needs reorganization:

* The results section should begin by enumerating the cost categories that emerged from the qualitative analysis, including their meaning as it might not be clear to everyone (e.g. what is transport cost?)

* Cost results should be grouped into standard categories, namely “fixed” and “variable”. For example, workload is a variable cost.

* In general, the results section seems disorganized. Results seem to appear out of nowhere and therefore it is hard to read. In addition to describing cost categories, the quantitative section should begin by outlining how results are presented.

* Considering rephrasing this sentence because it is not clear: “The overall number of CL lesions needing diagnosis is 1.7 to 2.2 times the total number of reported CL persons.”

* It should be clearly stated what the numbers inside brackets mean. Range? Minimum and maximum?

* This phrase is wrong: “ The p-values represent the healthcare professionals' (HP) general acceptability of the two diagnostic tools, calculated using a unidirectional ANOVA with Fisher's classic test.” That is not what a p-value is. Please explain or restate your affirmation.

* The regression analysis needs more justification, including an explanation of the variables included in the right-hand side of the equation.

* It is not surprising that the R-squared of the linear regression model is so high, given that the model is decomposing total cost into its constituent parts. Therefore, the high R-squared is an artifact of the data. Contribution of individual cost components is influenced by a) the share of each cost category, and b) how much variation was reported by HP.

* The scenarios are poorly explained and seem to appear out of nowhere in the results section. It is not enough to reference Figure 2, please explain what those scenarios mean in the body of the text.

* Qualitative results: really good. Uncommon in costing literature and it really adds value to the paper.

* Conclusion section should summarize the cost analysis, i.e. average cost for RDT and microscopy.

* Table 2: should read “interval” rather than “interval”, and “min” instead of “inf”, “max” instead of “sup”

* Table 3: needs better footnotes, i.e. what does SD mean? not clear for everyone that it stands for standard deviation. Also, not clear what the sample sizes represent e.g. 34 vs 6?

* Table 4:

* Do authors mean “standard error” instead of “standard deviation”?

* In a multivariate regression model, the intercept is not the base cost. Also, the footnote doesn’t match the table (48.2 MAD vs 52.3)

* R-squared is not a measure of model robustness.

6. PLOS authors have the option to publish the peer review history of their article (what does this mean?). If published, this will include your full peer review and any attached files.

**Do you want your identity to be public for this peer review?** For information about this choice, including consent withdrawal, please see our Privacy Policy.

Reviewer #1: **Yes: **Dr. Angela Nyangore Migowa

Reviewer #2: **Yes: **David Contreras Loya

---

## [Editor Report · Decision Letter 1]

20 Feb 2024

Micro-costing from healthcare professional's perspective and acceptability of cutaneous leishmaniasis diagnostic tools in Morocco: A mixed-methods study

PGPH-D-23-01769R1

Dear Pr Bennis,

We are pleased to inform you that your manuscript 'Micro-costing from healthcare professional's perspective and acceptability of cutaneous leishmaniasis diagnostic tools in Morocco: A mixed-methods study' has been provisionally accepted for publication in PLOS Global Public Health.

Best regards,

Andrés F. Henao-Martínez, M.D.

Academic Editor